# Identification of the Major Protein Components of Human and Cow Saliva

**DOI:** 10.3390/ijms242316838

**Published:** 2023-11-28

**Authors:** Srinivas Akula, Charlotte Welinder, Zhirong Fu, Anna-Karin Olsson, Lars Hellman

**Affiliations:** 1Department of Cell and Molecular Biology, Uppsala University, The Biomedical Center, Box 596, SE-751 24 Uppsala, Sweden; srinivas.akula@icm.uu.se (S.A.); annazhirongfu@gmail.com (Z.F.); 2Department of Clinical Sciences Lund, Division of Mass Spectrometry, Lund University, SE-221 00 Lund, Sweden; charlotte.welinder@med.lu.se; 3Department of Medical Biochemistry and Microbiology, The Biomedical Center, Box 582, SE-751 23 Uppsala, Sweden; anna-karin.olsson@imbim.uu.se

**Keywords:** saliva, IgA, BSP30, PIGR, odorant protein, mucin

## Abstract

Cows produce saliva in very large quantities to lubricate and facilitate food processing. Estimates indicate an amount of 50–150 L per day. Human saliva has previously been found to contain numerous antibacterial components, such as lysozyme, histatins, members of the S-100 family and lactoferrin, to limit pathogen colonization. Cows depend on a complex microbial community in their digestive system for food digestion. Our aim here was to analyze how this would influence the content of their saliva. We therefore sampled saliva from five humans and both nose secretions and saliva from six cows and separated the saliva on SDS-PAGE gradient gels and analyzed the major protein bands with LC-MS/MS. The cow saliva was found to be dominated by a few major proteins only, carbonic anhydrase 6, a pH-stabilizing enzyme and the short palate, lung and nasal epithelium carcinoma-associated protein 2A (SPLUNC2A), also named bovine salivary protein 30 kDa (BSP30) or BPIFA2B. This latter protein has been proposed to play a role in local antibacterial response by binding bacterial lipopolysaccharides (LPSs) and inhibiting bacterial growth but may instead, according to more recent data, primarily have surfactant activity. Numerous peptide fragments of mucin-5B were also detected in different regions of the gel in the MS analysis. Interestingly, no major band on gel was detected representing any of the antibacterial proteins, indicating that cows may produce them at very low levels that do not harm the microbial flora of their digestive system. The nose secretions of the cows primarily contained the odorant protein, a protein thought to be involved in enhancing the sense of smell of the olfactory receptors and the possibility of quickly sensing potential poisonous food components. High levels of secretory IgA were also found in one sample of cow mouth drippings, indicating a strong upregulation during an infection. The human saliva was more complex, containing secretory IgA, amylase, carbonic anhydrase 6, lysozyme, histatins and a number of other less abundant proteins, indicating a major difference to the saliva of cows that show very low levels of antibacterial components, most likely to not harm the microbial flora of the rumen.

## 1. Introduction

Ruminants have a very complex digestive system to facilitate the use of cellulose-rich food. However, they cannot process cellulose themselves due to the lack of an enzyme, a cellulase, that can separate the individual sugar units of cellulose for further use as a food source. To our knowledge no mammal has a gene for a cellulase, but cellulases are found in other parts of the animal kingdom [1]. Cows therefore need the help of a complex flora of microorganisms to process the cellulose-rich food and transform the energy of cellulose into macromolecules that are digestible for their intestinal enzymes that can be transported into the blood circulation by their transport receptors. Cellulose is a very stable molecule, and the processing of cellulose-rich food is therefore challenging. The ruminant digestive tract is therefore considerably more complex than in most other mammals. Cows have four stomachs, compared to only one in primates. In the first of them, the rumen bacteria and unicellular eukaryotes process the incoming food into microbial biomass via fermentation. To enhance this process, the rumen content is actively mixed by rumination, recurrently transported up to the mouth, chewed and then returned to the rumen. This involves lubrication by large amounts of saliva. Estimates have indicated that cows produce 50–150 L of saliva per day [2]. Saliva is known to contain a large and diverse set of proteins which perform multiple functions such as taste and digestion, lubrication, pH buffering and maintenance of general health by controlling the oral microbiota.

Human saliva has been shown to be complex mix of different proteins, including mucins 5B and 7, amylase, secretory IgA, carbonic anhydrase 6, lysozyme, lactoferrin, histatins, cystatins and many more [3]. Using peptide fractionation and proteomics, more than 2000 proteins have been identified in human saliva [4]. However, other studies have shown that nearly 98% of the total salivary protein is found in approximately 10 major protein bands on Coomassie stained gels, indicating that the majority of proteins identified by mass spectrometry are found is very low amounts [5,6]. Due to the low amounts, the majority of them most likely have little or no biological significance; however, they may still potentially serve as diagnostic markers [4]. The proteins of human saliva primarily originate from three salivary glands, the parotid, submandibular and sublingual salivary glands, but some material may also come from other tissues through plasma contribution [3]. Similarly, cow saliva is produced by a set of different salivary glands, but the amount of saliva differs markedly between these two mammalian species, which may influence the protein composition of the saliva.

Due to the major differences in the digestive system between humans and ruminants, we became interested in how this influences the protein components of the saliva between humans and cows. To further investigate this issue, we have separated cow saliva and mucus from the nose on SDS-PAGE gels, both native and deglycosylated samples, to identify the major components of cow saliva and to obtain a picture of the extent of carbohydrate at the individual components. We then analyzed major protein bands using liquid chromatography–tandem mass spectrometry (LC-MS/MS) to obtain information concerning the identity of the different protein bands. As reference material, we also sampled saliva from five human subjects and analyzed these samples with SDS-PAGE and MS analysis of major protein bands. The result primarily showed large differences in the amounts of the different components. Cow saliva contains very high amounts of only a few components, including the lubricating mucin-5B, carbonic anhydrase 6, a pH-stabilizing enzyme and the short palate, lung and nasal epithelium carcinoma-associated protein 2A (SPLUNC2A), also named bovine salivary protein 30 kDa (BSP30) or BPI-fold-containing family A member 2B, BPIFA2B [7]. In contrast, human saliva was more complex and contained large amounts of alpha amylase, which was not found in the cow saliva. Numerous antibacterial components have previously also been found in human saliva, indicating larger involvement in human saliva of antibacterial components to control the microbiome in the mouth. Cow saliva seems instead to have very low amounts of antibacterial components in order to not harm the microbial community in the rumen. Indications from drippings of one cow also indicates the potent upregulation of IgA, probably primarily in the nose as an alternative and effective protection against infection in cows.

## 2. Results

### 2.1. Sample Collection of Cow and Human Saliva

The project was started by obtaining a sample of drippings from the mouth of one cow from a commercial farm outside of Uppsala. Later, this was found not to be the optimal sampling procedure as drippings contain both material from saliva and the nose. This sample was divided into two tubes, and one of the samples was de-glycosylated by the addition of a mix of carbohydrate cleaving enzymes. This sample and the untreated sample were analyzed using SDS-PAGE (Figure 1). The major bands were extracted from the gel and analyzed using mass spectrometry (MS).

In order to determine variation between cows, we then ordered three new samples from three different cows from the same commercial farm. To our surprise, the SDS-PAGE pattern now looked very different between cows and also between two samples from the same cow. The reason for this large difference between samples was later shown to be the different amounts of material from the mouth and nose. We therefore contacted a large farm outside of Uppsala to obtain additional samples where we could better control the origin of the sample. We acquired samples from three individual cows. We took samples from both the mouth and the nose. The three individual cows now showed a very similar pattern for both their saliva and nose secretions, respectively. The protein patterns of saliva and nose secretions were, however, very different. For a comparative study between cow and human saliva, saliva samples were also obtained from five different humans.

Due to the complex pattern of sampling, the three cow samplings and the human samples will be described in separate sections.

### 2.2. SDS-PAGE Separation of Mouth Drippings from One Cow

Mouth drippings from one cow, cow A, were separated using SDS-PAGE under reducing conditions. Different volumes of saliva were used to obtain a suitable amount of protein for both good separation and for the subsequent identification of protein using LC-MS/MS analysis. In order to obtain information concerning the carbohydrate content of the different saliva proteins, one of the samples was treated with a mix of deglycosylation enzymes. After staining with colloidal Coomassie, eight dominating bands were cut out from the gel (Figure 1) and analyzed using LC-MS/MS.

### 2.3. LC-MS/MS Analysis of Individual Protein Bands from the SDS-PAGE Separation

The eight gel bands were enzymatically cleaved with trypsin, the generated tryptic peptides were analyzed using LC-MS/MS and the raw MS files were searched against a bovine database using Proteome Discoverer 3.0.

Bands 1, 2 and 8 were present in too low amounts to give conclusive results in the LC-MS/MS analysis. Bands 3, 4 and 6 were identified as the three components of secretory IgA. Band 3 was found to be the secretory component that is part of the transport receptor for IgA over epithelial layers, band 4 the IgA heavy chain and band 6 the immunoglobulin light chain. The protein of band 3, the secretory component, is part of the poly-Ig receptor named PIGR [8,9]. Band 5 appeared to be carbonic anhydrase VI, a salivary protein involved in the reversible hydration of CO_2_ that has been suggested to be involved in maintenance of pH homeostasis on tooth surfaces and of the mucosa of the gastrointestinal canal [10,11]. Band 7 was found to be the odorant protein, a protein that is thought to be involved in sensing smell by binding to olfactory receptors and enhancing odor sensing. The odorant-binding protein is a soluble dimeric protein with subunits of approximately 19 kDa, which fits nicely with the size on the gel (Figure 1) [12].

When examining at the peptides that appear in all of the eight bands, we found that five out of eight bands contained peptides originating from mucin-5B, the major salivary mucin. Three mucin-5B peptides were found in band 8, three in band 5, one in band 3, thirteen in band 2 and 30 in band 1, indicating an ongoing degradation of the mucin in the saliva. This shows the presence of high amounts of this mucin in the saliva, despite the fact that it does not enter the gel due to its large size and stains poorly due to its very high carbohydrate content. We also screened for antibacterial proteins, but only one peptide was found for lysozyme, and that was in band 8, with some lactoferrin in band 3 and some S100A8 in band 8, but they were not dominating.

### 2.4. Carbohydrate Content of the Salivary Components

Following the identification of several of the major bands, we went back to the gel analysis in Figure 1 to investigate the carbohydrate content of the various salivary proteins. As can be seen from the figure, both the secretory component (PIGR) and the IgA heavy chain are relatively heavily glycosylated, whereas neither cattle immunoglobulin light chains nor the odorant proteins seem to be glycosylated to any significant degree (Figure 1). Carbonic anhydrase 6 is also glycosylated as a drop in molecular weight from approximately 47 kDa to approximately 42 kDa on gel was observed (Figure 1).

### 2.5. Analysis of the Protein Bands in the Second Sampling of Drippings from Three Cows from the Commercial Farm

New samples from three individual cows, cows A, B and C, were ordered from the same commercial farm as in Figure 1. Mouth drippings from these three cows were analyzed using SDS-PAGE (Figure 2). To our surprise, the pattern looked very different between cows and also between the two samples from the same cow, cow A, when taken at different times (Figure 2). The second sample from cow A contained almost no IgA, indicating that at the first sampling cow IgA had an infection that resulted in a marked increase in IgA production. When we examined the samples from cows B and C, we could see that they showed very different patterns indicating a mix of content from the mouth and nose. The drippings from cow B had almost only content from saliva, with bands only for the approximately 50 kDa, which is carbonic anhydrase 6, and the approximately 30 kDa BPIFA2B (also named SPLUNC2A or BSP30) [13]. In contrast, cow C had the majority from the nose where the odorant protein is dominating. The second sampling of cow A showed an almost 50:50 mix of saliva and nose content but, as discussed above, very low levels of IgA and of the secretory component of IgA (Figure 2). Due to the large variation between cows and also between the same cow sampled at different time points, we wanted to be able to have better control of the sampling, which is why we contacted a farm to obtain samples from their cows under more controlled conditions. The results from these new samples are presented in the next section.

### 2.6. Analysis of the Protein Bands from Saliva and Nose of Three Cows

Careful sampling of saliva and nose secretions of three cows from a farm west of Uppsala resulted in very consistent results (Figure 3). SDS-PAGE analysis of these samples showed that the saliva contained primarily carbonic anhydrase 6 (band 1) and of BPIFA2B (band 2), whereas the nose secretion instead was dominated by the approximately 19 kDa odorant protein in band 4 (Figure 3). Both saliva and nose secretion had low levels of IgA and of the secretory component. However, the IgA bands were more pronounced in the nose and varied considerably between individuals (Figure 3). We can, for example, see that cow number 3 had considerably higher IgA levels than both cows 1 and 2 (Figure 3). The saliva from cow 1 seemed to be pure saliva, whereas saliva from both cows 2 and 3 most likely had a minor contaminant from the nose as there we could also see a minor band of the odorant protein (Figure 3). Band 5 was identified as prolactin-inducible protein homolog. Traces of a secretoglobin family 1D member was also detected in this band [14,15].

### 2.7. Analysis of the Protein Bands from Human Saliva of Five Different Persons

Saliva from five different persons were analyzed using SDS-PAGE (Figure 4). In contrast, from what we had experienced from the analysis of the cows, the samples looked very similar, with only minor variations in the protein bands between these five samples (Figure 4). Eight bands were excised from the gel and analyzed with LC-MS. Band 1 was found to be the polymeric immunoglobulin receptor (PIGR) with a molecular weight of 83.2 kDa (Figure 4). Band 2 was found to be albumin. Bands 3 and 4 was found to be alpha amylase 1B with a molecular weight of 57.7 kDa (Figure 4). Small amounts of the heavy chain of IgA also probably hid in one or both of these bands as we found the PIGR, which is directly bound to IgA in band 1 (Figure 4). Band 5 is most likely the 27 kDa BPI-fold-containing family A member 2B (BPIFA2B), also named SPLUNC2, which we found in high amounts in the bovine saliva sample (Figure 3 and Figure 4). This band was only seen in one of the five human samples, sample E. Band 6 is most likely also BPIFA2B. In band 7, we found peptides from the immunoglobulin light chain, which has a molecular weight of 23.4 kDa. In band 8, we found peptides from the prolactin inducible protein with a molecular weight of 16.6 kDa and of lipcalin-1 with a molecular weight of 19.2 kDa (Figure 4). In band 9, we found peptides for cystatin SN and cystatin SA, which both have molecular weights of 16.4 kDa. In band 10, we found peptides from histatin-1, which has a molecular weight of 7 kDa.

## 3. Discussion

The analysis of the cow and human saliva presented here gives strong indications that cow saliva has a less complex proteome than human saliva, at least when it comes to the major components. The major protein components of human saliva have been found to be amylase, histatins, IgA and mucin 5B, which is so large that it does not enter the gel, and an array of other more or less abundant proteins, including lysozyme and lactoferrin [16,17]. In contrast, bovine saliva seems to be dominated by three proteins, carbonic anhydrase 6, BPIFA2B and mucin 5B. Only a few peptides for other antibacterial proteins, including lysozyme, lactoferrin and histatins, were found to be minor components of some bands, indicating that these antibacterial compounds are found in relatively low amounts in cow saliva, possibly so as to not to interfere with the microbiome of the rumen. However, we see very high amounts of one potentially anti-microbial protein, BPIFA2B. This very abundant protein has been proposed to play a role in local antibacterial response by binding bacterial lipopolysaccharides (LPSs) and inhibiting bacterial growth [13]. However, recently, the role of this protein in bacterial defense has been questioned [18]. The protein may instead have a major function as a surfactant to facilitate the rumination by its surface activity to quickly wet the digested material for efficient degradation by enzymes of the microbial flora of the rumen [18]. Mucin 5B is a highly glycosylated protein with potent lubricating functions but may also inhibit some bacteria from adhering to teeth enamel and the mouth tissue and thereby affect bacterial colonization [19]. The third major protein of the cow saliva, carbonic anhydrase 6, has a function in turning CO_2_ into carbonate and thereby regulating pH in the oral cavity [10,11]. Carbonic anhydrase 6 is a member of a small family of related enzymes where this particular enzyme seems to be expressed primarily in the secretory glands producing saliva, thereby having a tissue-specific function to protect the enamel of the teeth by contributing to keep a favorable neutral pH in the oral cavity. This enzyme may have an especially important role for ruminants due to the large quantities of food ingested by the cows and the importance of keeping the microbial flora at near neutral pH for stable cellulose processing. These three proteins seem to make up the majority of total proteins in saliva, and all three have important functions in the process or rumination via pH stabilization, lubrication and acting as surfactants to enhance accessibility of enzymes to the digested material. The control of the bacterial flora of the mouth seems to a lesser extent to be carried out by the antibacterial substances that are found in human saliva such as lysozyme, lactoferrin, defensins, histatins and s100 members but rather by mucin 5B and IgA, which may inhibit attachment of bacteria to mucosal and dental surfaces as we observed a very strong upregulation of IgA in one of the cows during the first sampling (Figure 1). Low levels of secretory IgA were identified in most individuals, and the amounts between individuals and at different timepoints of sampling seemed to vary considerably, indicating a strong effect on the amounts produced depending on infection status (Figure 1, Figure 2 and Figure 3). The first sample we analyzed contained very high amounts of secretory IgA, and the second sampling from the same cow showed very low levels, indicating that they return to low baseline levels when an infection has been cleared (Figure 1 and Figure 2). This is our interpretation as we do not have any definite proof of the infection status of this cow at the first time of sampling. However, the dramatically higher levels of secretory IgA in that sample give a strong indication for this scenario. One interesting possibility is that this IgA can also help in forming and protecting the commensal microbiota composition of the rumen of cows [20]. An analysis has been performed to investigate the specificity of the IgA in cow saliva [20]. Interestingly, the result shows a relatively broad specificity of these antibodies to the bacterial composition of the rumen and not to the bacterial flora of the mouth [20]. We also found low levels of a few other proteins, including prolactin-inducible protein homolog a protein and the secretoglobin family 1D member, a glycoprotein of the lipophilin family, a protein relatively widely expressed in normal tissues, with not yet well-defined functions. However, they may have the ability to bind androgens and other steroids [14,15].

The protein components of the nose secretions were quite different from the saliva. Here, we have one very dominating component, the odorant protein, which is thought to be involved in odor sensing and thereby most likely an important component in the sensing of eatable and toxic plants as a food source for the cows (Figure 3). It has previously been found in nasal glands and secretions but not in saliva [12]. The function of the odorant proteins is not fully known but is thought to enhance the possibility of the olfactory receptors to sense pheromones and different odors and thereby have possible functions in partner selection and enhancing the capacity to avoid poisonous plants. They seem to have a major function primarily in the perception of substances that have low solubility in water, where a carrier protein may be needed to enhance transport to the receptors [21]. In vertebrates, the odorant proteins belong to the large lipocalin family. Members of this family have molecular weights spanning from 19 to 23 kDa [22,23]. However, odorant- and pheromone-binding proteins are found in animals as different as insects and mammals and derive from a number of different protein families with very different primary structures [24]. Interestingly the bovine salivary odorant proteins seem to be very homogenous in size, having only one molecular weight of around 19 kDa and no carbohydrate content as the molecular weight did not change upon treatment with the deglycosylation enzymes (Figure 1). They were found to be homodimers of a size of approximately 40 kDa, are produced by nasal glands and constitute approximately 1–2% of the protein content of nasal mucus [12,22,25]. However, our results indicate a much higher percentage of nasal mucus (Figure 1, Figure 2 and Figure 3). Interestingly, the odorant protein is also expressed in both the trachea and bronchi, indicating that it may have additional functions in addition to odor perception. We found very high levels of this protein in secretions of the nose, in agreement with previous reports where it was primarily found in the nasal olfactory and respiratory mucosa and in tears but not in saliva [12].

In conclusion, the protein content of saliva and nose secretions is apparently very different, and the protein content of human and cow saliva also show major differences, primarily in the amounts of the components. The large differences in numbers of the proteins between cow and human saliva most likely reflect the very large differences in amounts of saliva produced between these two mammalian species and the differences in the role that saliva have in their food intake, which for the cows is an adaptation to rumination so as to not harm the microbial community of the rumen.

## 4. Materials and Methods

### 4.1. SDS-PAGE Separation of Cow Salivary Proteins

Cow saliva was obtained from three cows at the commercial research farm outside of Uppsala Håtunalab (Uppsala, Sweden) and from three other cows at a regular farm also outside of Uppsala, six different cows in total. Samples of the cow saliva were mixed with 4× sample buffer, containing sodium dodecyl sulfate (SDS) (Invitrogen, Carlsbad, CA, USA). After addition of β-mercapto-ethanol to a final concentration of approximately 5%, the sample was mixed and heated to 85 °C for 5 min. These samples were then separated by gel electrophoresis on 4–12% pre-cast SDS-PAGE gels (Invitrogen, Carlsbad, CA, USA). Overnight staining in colloidal Coomassie staining solution followed by de-staining by several washes enabled the visualization of the protein bands [26].

In order to obtain information concerning the carbohydrate content of the different saliva proteins, one sample of the saliva was treated with a potent combination of deglycosylation enzymes using the most effective such deglycosylation mix on the market, the Biolabs deglycosylation mix II (New England Biolabs, Ipswich, MA, USA (P6044S)).

### 4.2. Analysis of Major Gel Bands from the SDS-PAGE Separation with LC-MS/MS

After staining and de-staining of the gels, prominent protein bands were excised from the SDS-PAGE and digested with trypsin, followed by identification using LC–MS/MS). Briefly, gel bands were washed with 400 µL MQ for 30 min on a shaker at RT, and the liquid was removed (after each of the following steps, the liquid was removed). The gel band was washed with 300 µL 40% acetonitrile in 25 mM ammonium bicarbonate 15–30 min, repeated twice. Then, 200 µL 100% acetonitrile was added to the gel bands and allowed to stand 5 min; this decreased the drying time for the gel bands. The gel bands were dried using a Speed Vac vacuum centrifuge (approx. 10 min). Reduction using 10 mM DTT at 56 °C for 30 min was followed by alkylation in 20 mM iodoactetamide for 30 min in darkness at room temperature. The gel pieces were washed and dried again before digestion with 20 µL of 0.04 µg/µL trypsin (Sequencing Grade Modified Trypsin, Part No. V511A) at 37 °C overnight (Promega, Madison, WI, USA). Peptides were extracted by the addition of 100 µL 1% formic acid in MQ for 10 min. The liquid was transferred to a new collection tube. To the gel band 100 µL, 100% acetonitrile was added, and the liquid was transferred to the collection tube after 5 min. These two last steps were repeated once. Then, the extracted peptides were speed vacuumed until dry and resolved in 20 µL 2% acetonitrile in 0.1% trifluoroacetic acid.

### 4.3. Mass Spectrometry Acquisition

The LC-MS/MS detection was performed on Tribrid Fusion mass spectrometer equipped with a Nanospray Flex ion source and coupled with an EASY-nLC 1000 ultrahigh-pressure liquid chromatography (UHPLC) pump (Thermo Fischer Scientific, Waltham, MA, USA). Peptides were injected into the LC-MS device. Peptides were concentrated on an Acclaim PepMap 100 C18 precolumn (75 μm × 2 cm, Thermo Scientific, Waltham, MA, USA) and then separated on an Acclaim PepMap RSLC column (75 μm × 25 cm, nanoViper, C18, 2 μm, 100 Å) at a temperature of 40 °C and with a flow rate of 300 nL/min. Solvent A (0.1% formic acid in water) and solvent B (0.1% formic acid in acetonitrile) were used to create a nonlinear gradient to elute the peptides. For the gradient, the percentage of solvent B was maintained at 3% for 3 min, increased from 3% to 25% for 60 min, increased to 60% for 10 min, increased to 90% for 2 min and then kept at 90% for another 8 min to wash the column.

The Orbitrap Fusion was operated in the positive data-dependent acquisition (DDA) mode. The peptides were introduced into the LC-MS device via a stainless steel nano-bore emitter (OD 150 µm, ID 30 µm) with the spray voltage of 2 kV and a capillary temperature of 275 °C. Full MS survey scans from *m*/*z* 350–1350 with a resolution of 120,000 were performed in the Orbitrap detector. The automatic gain control (AGC) target was set to 4 × 10^5^ with an injection time of 50 ms. The most intense ions (up to 20) with charge states 2–5 from the full scan MS were selected for fragmentation in the Orbitrap. The precursors in the second analyzer were isolated with a quadrupole mass filter set to a width of 1.2 *m*/*z*. Precursors were fragmented by high-energy collision dissociation (HCD) at a normalized collision energy (NCE) of 30%. The resolution was fixed at 30,000, and for the MS/MS scans the values for the AGC target and injection time were 5 × 10^4^ and 54 ms, respectively. The duration of dynamic exclusion was set to 45 s, and the mass tolerance window was 10 ppm.

### 4.4. Data Analysis

The raw files from LC-MS/MS were analyzed with Proteome Discoverer 2.5 (Thermo Scientific™) against the UniProt Bovine database (UP000009136) and bovine immunoglobulins and mucin 5B (GeneID = 789503) manually downloaded from https://www.ncbi.nlm.nih.gov/protein/ (6 February 2023) with the search terms bovine + immunoglobulin + mucin 5B. The precursor tolerance and fragment tolerance were set to 10 ppm and 0.05 Da, respectively. Trypsin was selected as enzyme, methionine oxidation and N-terminal acetylation were treated as dynamic modification and carbamidomethylation of cysteine as a fixed modification. Extracted peptides were used to identify and quantify them with label-free relative quantification. The extracted chromatographic intensities were used to compare peptide abundance across the gel bands.

## Figures and Tables

**Figure 1 ijms-24-16838-f001:**
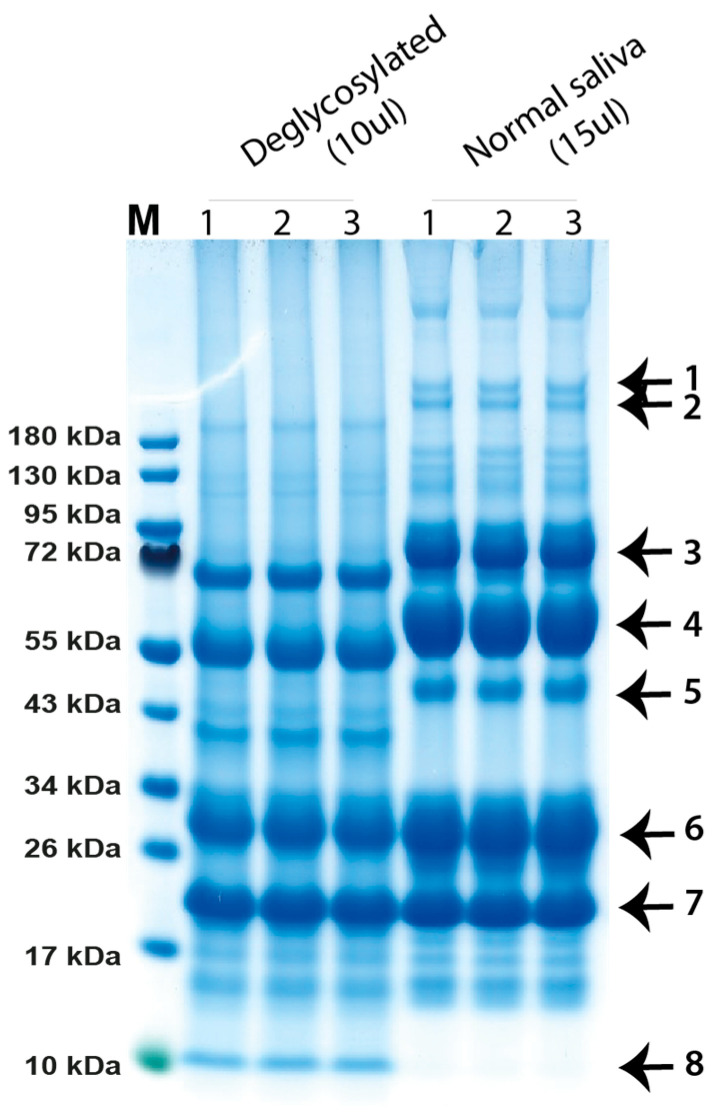
Separation of cow saliva on SDS-PAGE gels. Cow mouth drippings was sampled from one cow at the commercial veterinary farm Håtunalab located just outside of Uppsala in Sweden. The sample was transferred into two separate tubes where a combination of deglycosylation enzymes was added to one of the tubes and the sample was incubated overnight at 37 °C to remove the majority of carbohydrate chains. Following the overnight incubation of one of the tubes, sample buffer was added to both tubes and β-mercaptoethanol followed by heating to 85 °C for 4 min to denature the protein and break cysteine bridges for a better separation based only on the size on the SDS-PAGE gel. A number of lanes for both samples were loaded to obtain a sufficient amount of well separated protein for the LC-MS/MS analysis. Five major bands and three minor that seemed particularly interesting were excised for further analysis with MS (marked by arrows).

**Figure 2 ijms-24-16838-f002:**
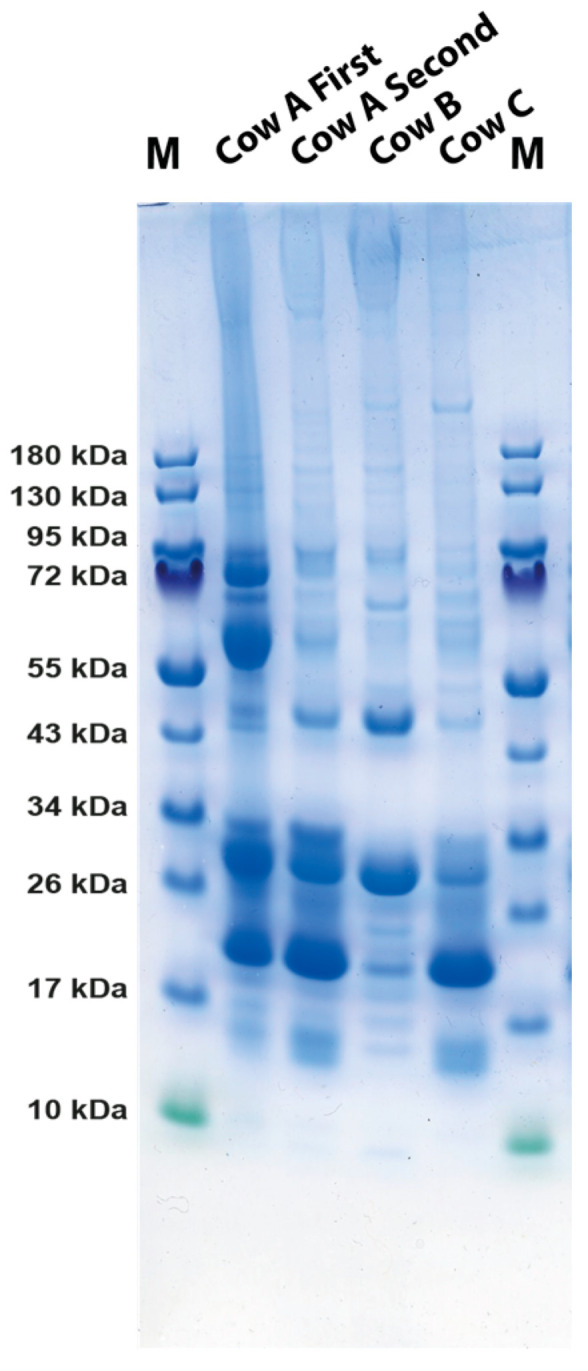
Separation of cow saliva on SDS-PAGE gels. Cow saliva from three different cows, cows A, B and C, from the commercial veterinary farm Håtunalab were separated on 4–12% PAGE gradient gels and stained with colloidal Coomassie brilliant blue. The sample from Figure 1 (normal saliva) was used as reference (cow A first). As can be seen from the figure, the variation between cows and from one sampling to another is considerable.

**Figure 3 ijms-24-16838-f003:**
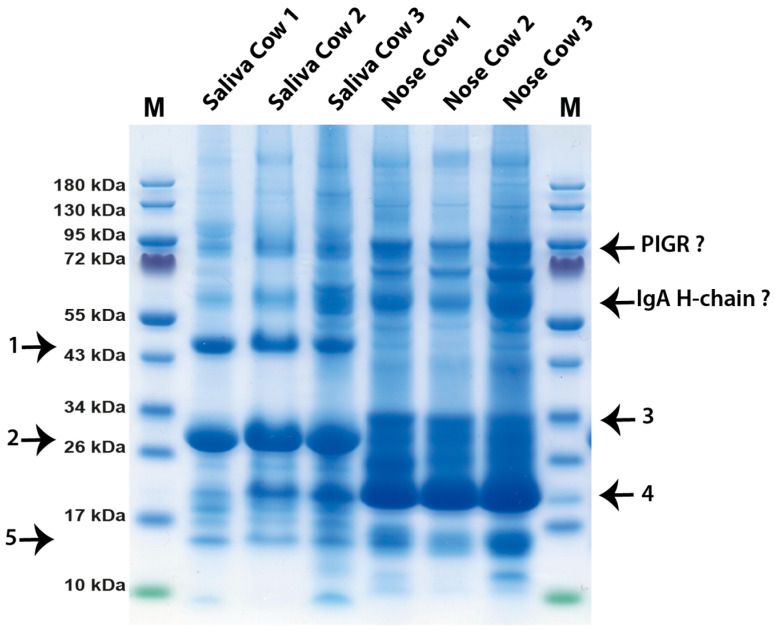
Separation of cow saliva and mucus from the nose on SDS-PAGE gels. Cow saliva and mucus from the nose from three different cows, cows 1, 2 and 3, were separated on 4–12% PAGE gradient gels and stained with colloidal Coomassie brilliant blue. Five different bands from this gel were cut out from the gel and sent for LC-MS analysis. These bands are marked by arrows and numbered from 1 to 5.

**Figure 4 ijms-24-16838-f004:**
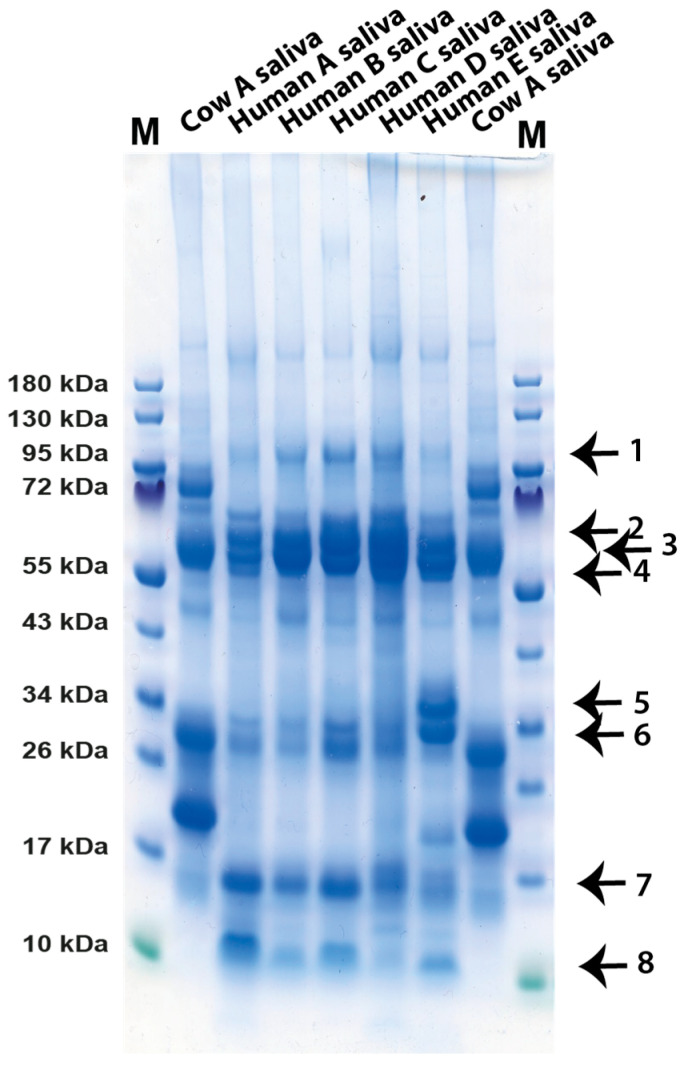
Analysis of human saliva from five different individuals. Human saliva samples from five different persons were analyzed on 4–12% PAGE gradient gels and stained with colloidal Coomassie brilliant blue. The cow saliva sample from Figure 1 (normal saliva) was used as reference (cow A first). Eight different bands from this gel were cut out from the gel and sent for LC-MS analysis. These bands are marked by arrows and numbered from 1 to 8.

## Data Availability

Data are contained within the article.

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
