# Peer review of "Identification of the Major Protein Components of Human and Cow Saliva"

_ijms, 2023, doi:10.3390/ijms242316838_

Round 1

Reviewer 1 Report

Comments and Suggestions for Authors

This manuscript has some interesting data but the presentation and interpretation is distracting.  The main question is why compare humans and cow saliva.  I think this could have been done to help establish methods (SDS-PAGE etc,) for which there is already a lot published on human saliva to which the authors should refer.  One of their discoveries is that muffins are difficult to stain, with Coomassie, which is why another stain Perioodic acid Schiff's is sometimes used.  This could then be applied to the cow saliva and it could confirm that the deglysocylation experiments have worked - a difference in apparent molecular weight is a secondary effect.  Perhaps a useful byproduct of using human saliva to establish methods is that you could (and do) compare the salivary proteome between man and cow.  In which case we need a more detailed analysis of the mass spec data, as opposed to just identifying the main bands which is also dependant not eh quality of the bovine saliva database- which might not be as complete as the human database.  Your discover that drippings may include nasal secretions is not that surprising (in the cow at least) but you describe the sequence of events in too much detail.  Again the differences in IgA concentrations could also be important but you haven't controlled collection conditions sufficiently to make claims about the health of that cow, so again the speculation is too much.  Its important to note that you have only presented concentrations and not total protein outputs of the saliva, so it is difficult to make comparisons between human and cow.  Perhaps you could do a timed sample? Overall the paper has potential but needs reworking and some more analysis before publication.

Abstract: the reasons to compare man and cow saliva are not clear, whilst "wonder" is a good start we need something more considered.

Fig 1: add a comment about the numbers and arrows- presumably proteins of interest? 

Section 2.3 presumably the muc5B degradation started before separation on gel, so its not necessary to say that the MUC 5b can't enter the gel as this refers to the whole molecule.

Section 2.5 too much speculation about the health of the cow at time of sampling relating to IgA concentration. 

Ln 169 i believe there is a new naming convention for PLUNCs, which you should use. 

More consistency in terms used- sometimes "drippings" sometimes "saliva".

Ln 196 provide ref for PIP involved in water regulation in glands. I think it unlikely as it is a s secretory protein. Also leave this till the discussion, it shouldn't be in the results. Poor writing style- sometimes in first person, sometimes third,  Given as a chronological order, don't need all the details about what you found next etc., 

eg the nasal secretions versus saliva story is obvious and doesn't require so much explanation. 

Comments on the Quality of English Language

Write in third person, remove speculation from results section, define the research question more clearly.  

Author Response

Reviewer 1.

We have now added several additional references concerning the human saliva in the introduction (Marked in red).

The problem with Mucin 5B is not mainly the staining but the size so that it is difficult to get it to enter the gel. However, as mentioned in the manuscript we find numerous peptides in almost all of the major bands on the gel indicating that degradation products of this protein and that it is abundant and that the original protein is found in relatively large amounts in the original sample.

Concerning the need for a more detailed analysis by mass spec we are skeptical as we know from years of experience that the high sensitivity the MS studies performed and the almost complete lack of quantitative information in these studies gives a very unrealistic picture of the proteome. We have sent purified recombinant protein bands to three different MS facilities where the major band we wanted to verify was present in around 85-95% of the total protein and we get from all three labs a list of 100-200 proteins and when we ask the providers if they can tell us which of the proteins that is our main protein they fail. I have therefore refused to pay the cost as the quality of the analysis is so bad.  In order to get a more accurate estimate of which proteins that actually exist in biologically relevant amounts we therefore have turned to SDS-gels and Coomassie staining as that gives us quantitative estimate of the most important components and also of the reproducibility between individuals.  We have now added several references on MS analysis of human saliva where more than 2000 proteins have been identified but where other authors clearly state that 90-99% of the total protein in saliva is found in approximately 10 bands on an SDS-PAGE gel indicating that the absolute majority of the total protein is found in a few very abundant proteins as we also see in this analysis. The reproducibility between individuals can also be seen from the five human samples and the three last cow samples where we could more accurately control the origin then they are almost identical, which shows the reproducibility and quality of the assay.

Concerning the very high amounts of IgA in one sample we do not make any claims about health but just that it is an interesting finding that needs to look into closer in the future as it may point towards a major difference between cows and humans that cows may rely more on a rapid IgA response as their content of antibacterial proteins in saliva and nose secretions are lower than humans. In the discussion we say  ¨ This is our interpretation as we do not have any definite proof of the infection status of this cow at the first time of sampling.¨ This to make readers aware that this is a single finding  and that it is our interpretation of this indication.

In the abstract we have now more clearly stated that the aim is to compare cow and human saliva in light of the major difference in food digestion between these two mammals (Marked in red).

We have now explained why we have selected these protein bands marked with arrows.

We have now modified according to the new PLUNC nomenclature also including the other names of this protein, BSP30 and BPIFA2B and the reference to this new name (Bingle CD., et al Biochemical Society Transactions. 2011; 39 (4): 977-83). We describe this in the abstract and introduction and use then BPIFA2B in the rest of the manuscript.

We have now updated the use of saliva and drippings to correctly apply to the sampling origin. We did however not find many that were incorrectly labeled.

The original PIP reference concerning water regulation could not be found, from the reference stating in another publication where it was referenced why the statement has been removed (Thanks for observing).

The nasal versus saliva is actually unexpected for us as when we sampled the nose it was almost impossible to get a sample after a lot of effort we got 20-50 ul whereas the saliva was very easy to get several milliliters so to us it was unexpected that we could see such a large influence by nose secretions on the drippings.

We have changed all text to third person and we have defined the research question and the conclusions more clearly in abstract, introduction and discussion (Marked in red).

Reviewer 2 Report

Comments and Suggestions for Authors

see attached file

Author Response

Reviewer 2.

We prefer to keep the text in the Intro to give the readers an overview of the experiments and why they have been made in that order. If this text is moved to the Materials and methods no one will read it and may have difficult to follow the different samplings and why they are performed in that order.

We have added several additional references concerning previous work on human saliva (Marked in red in the Introduction).

We have now clarified the aim of the study by changing the text of the abstract now marked in red.

Concerning the number of cows where we only have three where the sampling was made in a more accurate way. By looking at the three cows we can see that the protein pattern is almost identical indicating very little variation between cows why adding two three or more would only marginally affect the result. Essentially the only difference we see is a small variation in IgA levels between cows and that two of the cows have a low level of odorant protein indicating a minor contamination from nose secretion. The same high reproducibility is observed from the five human samples. Essentially only one of the samples have one or possibly two minor extra bands showing that both human and cow saliva show very low variation between individuals.

The need for a more accurate sampling procedure was not obvious to us before staring the project and is still puzzling due to the low amount of nose mucus we find. The commercial farm did not have permit to perform separate sampling why drippings was what we got. We realized, as can be seen from the first samples, that the large variation between samples was the varying amounts of nose to saliva in the sample. We feel these data are important for persons that want to look deeper into this issue as the sampling procedure is very important and that even the low amounts of nose secretion can massively influence and in a very uncontrollable way influence the result of the study. Why we want to keep this relatively long start of the results is to show the major influence on nose secretions in drippings but also the quite interesting finding of the high IgA content of one sample, the first, from cow A indicating a strong upregulation of IgA potentially as a response to infection, a finding we feel needs further studies in the future. If we had done a pre-sampling from only one cow we would not have detected the large variation in the protein content of the drippings, that was evident first after the two initial samplings of several cows.

Concerning the potential discussion within the results section. We have now removed these sentences or parts of sentences and in some cases added them to the discussion when it was lacking there.  

A more clear conclusion has been added to the end of the abstract and also to the end of the introduction and the discussion.

Round 2

Reviewer 1 Report

Comments and Suggestions for Authors

The authors have answered my points satisfactorily. 

Reviewer 2 Report

Comments and Suggestions for Authors

A scientific article should be written according to standard accepted rules. Authors should not place information in inappriopriate sections even if they are concerned that the reader will not read all of it.